# Strengthening of enterococcal biofilms by Esp

**Lindsey Spiegelman[1], Adrian Bahn-Suh[1¤a], Elizabeth T. Montaño[2], Ling Zhang[1¤b], Greg L. Hura[3], Kathryn A. Patras[2¤c], Amit Kumar[1], F. Akif Tezcan[1], Victor Nizet[2], Susan E. Tsutakawa[3], Partho Ghosh[1]***

**1** Department of Chemistry & Biochemistry, University of California, San Diego, La Jolla, California, United States of America, **2** Division of Host-Microbe Systems and Therapeutics, Department of Pediatrics, University of California, San Diego, La Jolla, California, United States of America, **3** Molecular Biophysics and Integrated Bioimaging Division, Lawrence Berkeley National Laboratory, Berkeley, California, United States of America

¤a Current address: Kaiser Permanente Bernard J. Tyson School of Medicine, Pasadena, California, United States of America
¤b Current address: Slingshot Biosciences, Emeryville, California, United States of America
¤c Current address: Department of Molecular Virology and Microbiology, Baylor College of Medicine, Houston, Texas, United States of America
* pghosh@ucsd.edu

**Data Availability Statement:** The structure was deposited in the RCSB Protein Data Bank (https://www.rcsb.org/structure/6ORI).

**Funding:** This work was supported by NIH grants T32 GM007240 (LS), R56 AI096837 (PG and VN),

## Abstract

Multidrug-resistant (MDR) *Enterococcus faecalis* are major causes of hospital-acquired infections. Numerous clinical strains of *E. faecalis* harbor a large pathogenicity island that encodes enterococcal surface protein (Esp), which is suggested to promote biofilm production and virulence, but this remains controversial. To resolve this issue, we characterized the Esp N-terminal region, the portion implicated in biofilm production. Small angle X-ray scattering indicated that the N-terminal region had a globular head, which consisted of two DEv-Ig domains as visualized by X-ray crystallography, followed by an extended tail. The N-terminal region was not required for biofilm production but instead significantly strengthened biofilms against mechanical or degradative disruption, greatly increasing retention of *Enterococcus* within biofilms. Biofilm strengthening required low pH, which resulted in Esp unfolding, aggregating, and forming amyloid-like structures. The pH threshold for biofilm strengthening depended on protein stability. A truncated fragment of the first DEv-Ig domain, plausibly generated by a host protease, was the least stable and sufficient to strengthen biofilms at pH $\leq$ 5.0, while the entire N-terminal region and intact Esp on the enterococcal surface was more stable and required a pH $\leq$ 4.3. These results suggested a virulence role of Esp in strengthening enterococcal biofilms in acidic abiotic or host environments.

## Author summary

The bacterium *Enterococcus faecalis* is part of the normal microbiome but can also cause serious hospital-acquired infections. *Enterococcus* strains isolated from hospitals tend to have certain proteins not found in microbiome strains. Such proteins are therefore likely to be important in infection. We sought to understand the function of one such protein,

R01 AI154149 (PG), and R01 GM137021 (SET and GLH); and DOE grant DE-SC0003844 (FAT and LZ). The funders had no role in study design, data collection and analysis, decision to publish, or preparation of the manuscript.

**Competing interests:** The authors have declared that no competing interests exist.

Esp, through biochemical, biophysical, and microbiological techniques. We found that Esp, which is on the bacterial surface, formed amyloid-like fibrils that prevented removal of biofilms. Biofilms are bacterial communities enmeshed within a matrix, and form within the body or on inert objects like catheters. They promote infection by increasing resistance to antibiotics and interfering with clearance by the immune system. We observed that biofilms that lacked Esp could be disrupted much more easily than those that had Esp. We also found that Esp acted only at low pH (i.e., acidic conditions). Exactly how low a pH depended on whether Esp remained on the bacterial surface or was liberated from the surface by a protease, with a human intestinal protease being a likely cause of liberation. In summary, we found that Esp acts at acidic conditions and likely contributes to virulence by preventing the dispersal of biofilms.

## Introduction

Multidrug-resistant (MDR) *Enterococcus faecalis* and *E. faecium* are major causes of life-threatening hospital-acquired infections [1]. A number of virulent strains of these gram-positive bacteria harbor a large pathogenicity island that encodes enterococcal surface protein (Esp). Examples include MDR *E. faecalis* MMH594 and vancomycin-resistant V586, and MDR patient isolates of the *E. faecium* CC17 clonal lineage [2–5]. Esp belongs to the "periscope" family of proteins [6], which are characterized by having a variable number of tandemly arranged C-terminal repeats that serve to structurally project a functional N-terminal non-repeat region outwards from the bacterial surface (Fig 1A) [2]. The C-terminal repeats constitute the majority of this ~200 kDa protein, while the N-terminal non-repeat region is ~75 kDa. Esp is covalently attached to the bacterial cell wall through an LPXTG-like motif at its C-terminus [2].

Esp is suggested to promote biofilm production [5, 7, 8]. Evidence for this function comes from experiments in which *esp* was deleted in a clinical *E. faecium* strain [5], or expressed from a plasmid in *E. faecalis* strains lacking *esp* (i.e., FA2-2 and OGR1F) [8]. In these experiments, the presence of *esp* resulted in greater biofilm mass, as measured by staining biofilms with crystal violet (CV), a non-specific dye. The *Enterococcus* biofilm matrix is composed of polysaccharides, proteins, and DNA [9], and enhances resistance to antibiotics and immune clearance [10]. Biofilms have roles not only in host niches but also on abiotic surfaces, such as catheters [11]. The Esp N-terminal non-repeat region, when expressed from a plasmid in the *esp*⁻ strain *E. faecalis* FA2-2, is sufficient for increased biofilm production [12]. A similar effect on biofilm production, albeit evaluated only qualitatively, is seen when a portion of the Esp N-terminal region is heterologously expressed from a plasmid in *Staphylococcus aureus* [13]. At low pH (4.2), this same portion of the Esp N-terminal domain forms an amyloid-like structure [13], which is not uncommon in bacterial biofilms [14, 15]. Amyloid-like fibrils are produced by numerous types of proteins and arise from unfolding of the native protein conformation and misfolding into a β-strand-rich structure.

However, other results suggest that Esp is not required for biofilm production. For example, a number of *E. faecalis* strains that lack *esp*, such as OGR1F, are fully capable of forming biofilms [16]. Not all strains that express *esp* produce biofilms [7, 17, 18], and expression of *E. faecalis esp* in an *E. faecium* strain lacking *esp* did not promote biofilm production [12]. In addition, Esp is seen to promote biofilm production only under the limited condition of glucose being present in the growth medium [5, 7, 8, 12, 13]. Indeed, whether *esp* contributes to

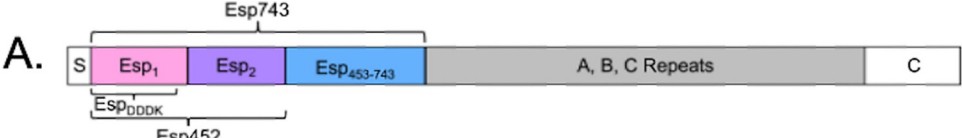

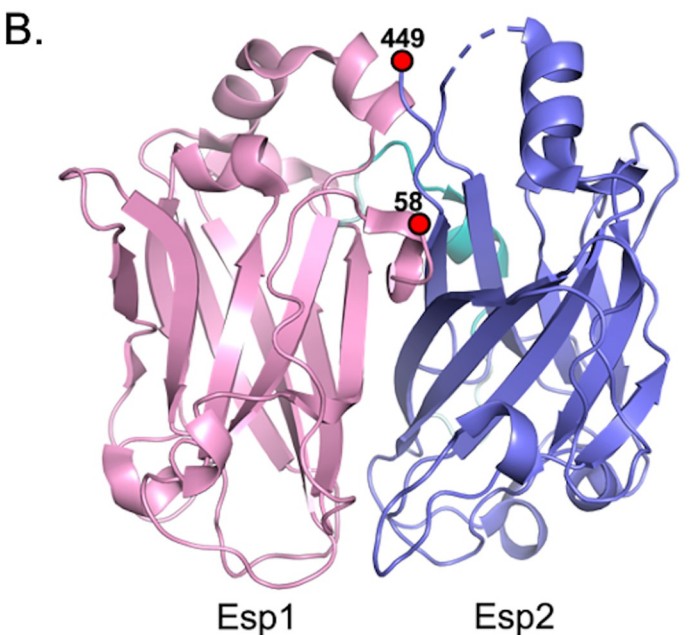

**Fig 1. Esp domains. A.** Schematic of Esp domains. The N-terminal non-repeat region, encompassed by Esp743 (and composed of Esp1, Esp2, and Esp$_{453-743}$ regions), is followed by A, B, C repeats and a C-terminal domain ("C"). The extent of the Esp$_{DDDK}$ and Esp$_{452}$ fragments is indicated, and "S" denotes the signal sequence. **B.** Structure of Esp452 in ribbon representation with Esp1 and Esp2 domains in pink and blue, respectively. The loop connecting the two domains is in cyan. Amino acids 407–419 lacked electron density and were not modeled (dashed line).

biofilm production by MMH594, the strain in which it was first identified [2], has not been reported.

The existence of Esp on a pathogenicity island and the high sequence conservation of its N-terminal non-repeat region among *E. faecalis* strains suggest a functional role for Esp. To gain further insight into Esp function, we pursued biochemical and structural studies of its ~75 kDa N-terminal region. We found that the N-terminal region was not required for biofilm production, but had a significant effect in strengthening enterococcal biofilms against mechanical or degradative disruption. Biofilm strengthening was dependent on low pH, which was brought about by fermentative catabolism of glucose in the medium. Low pH ($\leq 4.3$) promoted the unfolding of the Esp N-terminal region from its native conformation, resulting in aggregation and formation of an amyloid-like structure. Intact Esp on the enterococcal surface also required this low of a pH to strengthen biofilms. A ~20 kDa fragment, Esp$_{DDDK}$, from the very N-terminus of the mature form of Esp (i.e., after removal of its signal sequence), which could conceivably be produced by the host protease enteropeptidase, was sufficient for biofilm strengthening. This fragment was less stable than the intact N-terminal region, and strengthened biofilms at pH $\leq 5.0$. Our results suggest that Esp functions to strengthen enterococcal biofilms in acidic abiotic or host environments.

## Materials & methods

### Ethics statement

Animal experiments were approved by the UC San Diego Institutional Animal Care and Use Committee (IACUC) under protocol number S00227M, and conducted under accepted veterinary standards.

### Protein expression constructs

The coding sequences of Esp452 (aa 1–452, i.e., including the putative signal sequence) and Esp743 (aa 48–743) were amplified by PCR from *E. faecalis* MMH594 and inserted through restriction digestion and ligation into pET28a vector (Novagen) that was modified to encode a C-terminal PreScission protease cleavage site followed by a His$_6$-tag. The coding sequence of Esp$_{453-743}$ was amplified by PCR from the pET28a-Esp743 vector and inserted through restriction digestion and ligation into the modified pET28a vector. Expression constructs for Esp1 (aa 1–241) and Esp$_{DDDK}$ (aa 1–226) were generated by inverse PCR from the pET28a-Esp452 vector using the Agilent QuikChange II kit. The integrity of DNA constructs was confirmed by sequencing (Genewiz). Esp743 differed from other constructs in having Ala (encoded by the PCR primer) rather than Val at position 57. Esp isolates have Ala, Val, or Ile at this position, which occurs on a disordered portion of the protein (see below). Sequences of primers are in S1 Table.

### Expression and purification

Plasmids encoding various Esp regions were transformed into *Escherichia coli* BL21 (DE3) Gold. Transformed bacteria were grown with shaking at 37˚C in LB broth containing 50 μg/mL kanamycin to OD$_{600}$ 0.6–0.8, and then induced with 1 mM isopropyl β-d-1-thiogalactopyranoside (IPTG). Thereupon bacteria were grown overnight with shaking at 16˚C. For expression of selenomethionine (SeMet)-labeled Esp452, bacteria were grown and SeMet incorporated as previously described [19]. Bacteria were harvested by centrifugation (1,700 x *g*, 20 min, 4˚C), resuspended, and incubated for 15 min in lysis buffer (300 mM NaCl, 20 mM Tris, pH 8.0) supplemented with 1 mg/mL lysozyme and 1 mM PMSF. In the case of Esp743, the lysis buffer also included 20 units/μL DNase (Thermo EN0521), 2.5 mM MgCl$_2$, and 1 mM CaCl$_2$. Resuspended bacteria were lysed by sonication. The lysate was centrifuged (16,000 x *g*, 20 min, 4˚C) and the supernatant clarified through a 0.8 μm filter using a syringe. The filtered lysate was applied to a Ni$^{2+}$-NTA column that had been equilibrated with lysis buffer. In the case of Esp743, the lysate was incubated on the column for 20 min. The column was washed with five column volumes of lysis buffer, followed by five column volumes of wash buffer (lysis buffer + 0 mM imidazole for Esp452, and + 5 mM imidazole for all other fragments). Samples were then eluted with lysis buffer supplemented with 100 mM imidazole. After confirmation of purity by SDS-PAGE, eluted fractions were placed in dialysis tubing (6–8 kDa cutoff), with or without 25 μg/mL His$_6$-tagged PreScission protease in lysis buffer containing 2 mM DTT, and dialyzed overnight in the same buffer. When applicable, PreScission protease was removed by reverse-nickel chromatography. Esp constructs were then concentrated by ultrafiltration to 10–40 mg/mL ($\varepsilon_{280calc}$ 54,780 M$^{-1}$cm$^{-1}$ for Esp452; 69,220 for Esp743; 14,440 for Esp$_{453-743}$; 22,920 for Esp$_{DDDK}$; 24,410 for Esp1), filtered by syringe through a 0.8 μm filter, and applied to a Superdex 200 column (GE Healthcare) for size-exclusion chromatography in 150 mM NaCl, 20 mM Tris, pH 8.0. Fractions from the column were pooled and dialyzed into either 10 mM NaCl, 10 mM Tris, pH 8.0 for crystallization or phosphate buffered saline (PBS), pH 7.4 for all other experiments.

## Crystallization

Crystals of Esp452 were grown by the sitting drop diffusion method at 20˚C with drops composed of 1.0 µL Esp452 (7.5 mg/mL) and 1.0 µL precipitant solution containing 16% PEG 3350, 200 mM ammonium acetate, pH 7.0. For hydrogel polymer stabilization [20], crystals were soaked in the precipitant solution supplemented with 100 mM $CaCl_2$, and then soaked in a drop of precipitant solution containing 8.625% (w/v) sodium acrylate, 2.5% (w/v) acrylamide, and 0.2% (w/v) bis-acrylamide for 48 h. Crystals were transferred to a fresh drop of precipitant solution containing 1% ammonium persulfate and 1% TEMED for 10 min, and flash-cooled in liquid $N_2$.

Crystals of SeMet-labeled Esp452 and some crystals of native Esp452 were not stabilized in the hydrogel polymer. These particular crystals were grown at 20˚C with drops composed of 0.75 µL native or SeMet-labeled Esp452 (8 mg/mL) and 1.0 µL precipitant solution containing 20% PEG 3350, 200 mM ammonium citrate, pH 7.0. In the case of native Esp452, the drops also contained 12.5 mM $CaCl_2$, or in the case of SeMet-labeled Esp452, 3.75% sucrose. These crystals were cryoprotected by soaking in precipitant solution supplemented with 20% 2-methyl-2,4-pentanediol, and flash-cooled in liquid $N_2$.

## Structure determination

Anomalous dispersion data were collected from SeMet-labeled crystals of Esp452 to 2.3 Å resolution limit (Advanced Light Source beamline 12.3.1), and indexed, integrated, and scaled using HKL2000 (S2 Table, Esp–SeMet) [21]. Phases were determined with Autosol [22], which identified four anomalous scatterers per asymmetric unit. These corresponded to four of the six methionines of Esp452, with a single molecule of Esp452 occupying the asymmetric unit. The initial model was generated by automated building in Autosol and refined with phenix. refine using default parameters [22]. Amino acids and other molecules were modeled into electron density manually in Coot, as guided by inspection of $\sigma_A$ weighted $2mF_o\text{-}DF_c$ and $mF_o\text{-}DF_c$ maps, The model was then further refined against diffraction data of higher resolution limit, 2.1 Å (Advanced Photon Source beamline 24-ID-E), collected from crystals of native Esp452 that had been soaked in $CaCl_2$ (S2 Table, Esp + $Ca^{2+}$). Data were processed as described above, and phases were determined by molecular replacement using Phenix (MR-Phaser) with the model of Esp452 that had been refined against the Esp—SeMet data set. This model of Esp452 was further refined against the Esp + $Ca^{2+}$ data set.

This further refined model served as the molecular replacement phasing model for diffraction data collected from hydrogel polymer-stabilized Esp452 crystals, which diffracted to 1.4 Å resolution limit (Advanced Light Source beamline 5.0.2). These high-resolution data were scaled using Aimless [23], and phases determined by molecular replacement using MOLREP (S2 Table, Esp). Model building, refinement, and inspection of maps was carried out as described above. Electron density for the main chain in the final model was visible throughout, except for amino acids (aa) 50–58, 408–418, and 449–452. Waters were modeled using Coot and verified by manual inspection of difference maps. The final model contained a $Ca^{2+}$ ion and 433 waters. Reflections that were unique to the 1.4 Å resolution limit dataset constituted 69.3% of total reflections, and the $R_{work}$ and $R_{free}$ of these were 15.4% and 18.9%, respectively. The structure was deposited in the RCSB PDB (6ORI).

Pymol (The PyMOL Molecular Graphics System, Version 1.2r3pre, Schrödinger, LLC) was used for generating molecular figures.

## Fibrinogen binding

Human fibrinogen (Millipore Sigma) was resuspended in PBS at 1 ng/µL, and 100 µL was used to coat wells of a 96-well plate (Corning, #3603). The plate was incubated overnight at 4˚C.

Wells were washed 3X with TBST (150 mM NaCl, 20 mM Tris, pH 8.0, 0.1% Tween), then blocked with TBST + 0.1% bovine serum albumin (BSA) for 1 h at RT. The wells were washed 3X with TBST. Esp452-His$_6$ (3 μg), M1-His$_6$ (1 μg), or PBS was added to the plate, and the plate was nutated for 1.5 h at RT. The wells were then washed 3X with TBST. Anti-His HRP-conjugated monoclonal antibody was added at 1:500 dilution for 1 h at RT. The wells were washed 3X with TBST, and then TMB substrate was added as recommended by the manufacturer (BD Biosciences). The OD$_{450}$ was measured by plate reader (TECAN).

## Glycan screen

Binding of Esp452-His$_6$ to human glycans was assessed by the National Center for Functional Glycomics at Harvard University (https://ncfg.hms.harvard.edu/). These data are listed in S3 Table.

## UTI model

*E. faecalis* MMH594 and MMH594b (Δ*esp*) were grown overnight in BHI media to approximately 10$^9$ CFU/mL. Urine was voided from ten week-old C57bl6 mice (Jackson Laboratory) and 50 μL containing ~2 x 10$^8$ CFU were injected through the urethra under anesthesia as described previously [24]. At 1 day after inoculation, urine from each mouse was collected, diluted, and plated on BHI agar containing 20 μg/mL Erm. Five mice were sacrificed, and their kidneys and bladders homogenized and plated. This procedure was repeated on days 3 (two experiments) and 5 (one experiment). The experiment was performed twice independently and results combined for a total of ten mice/group on days 1 and 3 and five mice/group on day 5.

## Biofilms

*E. faecalis* was grown on plates containing Brain Heart Infusion (BHI) agar supplemented with antibiotics: MMH594, 20 μg/mL erythromycin (Erm); MMH594b (Δ*esp*), 20 μg/mL Erm and 20 μg/mL chloramphenicol (Cm); FA2-2 and OG1RF, 25 μg/mL rifampicin. Strains transformed with pEsp [8] were grown with 500 μg/mL spectinomycin in addition to the aforementioned antibiotics. Single colonies were picked from plates and grown in 5 mL shaking cultures in BHI containing antibiotics for 16 h, and diluted to an OD$_{600}$ of 2.7. Cm was omitted at this stage for MMH594b (Δ*esp*) because it slowed bacterial growth. This culture was inoculated 1:100 into 1.0 mL tryptic soy broth (TSB) containing 0.5% w/v glucose (TSBG) in a 12-well plate and grown at 37°C, with varying concentrations of Esp fragments or an equivalent volume of PBS for 19–20 h. The planktonic fraction of the culture was removed by pipette. The biofilms were washed twice with 500 μL TSB, and then resuspended in 500 μL 1.5 M NaCl. The resuspensions were centrifuged (4500 x *g*, 5 min, 4°C) and the supernatants discarded. Bacterial pellets were resuspended in 1.0 mL TSB, and 100 μL of each sample added to a well of a black, clear-bottom 96-well plate (Corning, #3603). Reconstituted CellTiter-Glo reagent, which contained lysis buffer and luciferase, was added to each well and samples were measured according to manufacturer's recommendations with a TECAN well-plate reader.

For DNase experiments, biofilms were grown as described above. At 19 h after inoculation, DNase II (Sigma) was added to a final concentration of 750 μg/mL. Biofilms were incubated for an additional 3 h at 37°C, and then washed and harvested as described above.

For measurements with crystal violet, biofilms were grown as described above but in a 96-well plate (Corning, #3603) with 100 μL media per well. Culture supernatants were removed and biofilms were washed 2X with 100 μL PBS. Biofilms were dried inverted for 1 h at RT, and then stained with 100 μL of 0.2% crystal violet for 20 min. The crystal violet solution

was removed, and the wells were washed 2X with 120 μL water. The stain was solubilized with 100 μL of 4:1 ethanol:acetone and vigorous pipetting. $OD_{595}$ of the ethanol:acetone mixture was measured with a TECAN plate reader.

### Thioflavin T

For *in vitro* experiments, 25 μg of Esp constructs were incubated in a 96-well plate in 100 mM sodium citrate buffer at pH's ranging from 4.2 to 6.0 and containing 20 μM Thioflavin T (Sigma). The plate was incubated at 37°C for 24 h, then shaken for 60 sec in a well-plate reader and fluorescence at 454 nm was measured. The optical density at 400 nm was measured immediately afterwards.

For biofilm experiments, Esp constructs were serially diluted in TSBG in wells of a 96-well plate (Corning, #3603). MMH594b (Δ*esp*) biofilms were grown in these wells as described above. The culture supernatant was aspirated, and biofilms were washed twice with 50 μL TSB. Thioflavin T was added to each well in sodium citrate, pH 4.5 and incubated for 10 min without shaking. Fluorescence was measured with a TECAN well plate reader.

### Confocal microscopy

Esp452 and Esp743 were covalently labeled with Alexa Fluor 647 (AF647) using Alexa Fluor 647 NHS Ester (Invitrogen), according to the manufacturer's recommendations. Biofilms were grown, as described above, in an 8-well chamber slide (NuncR Lab-Tek II, Thermo Scientific) containing 250 μL TSB with 0.5% glucose, 2.5 μL *E. faecalis* MMH594b (Δ*esp*) overnight culture ($OD_{600}$ 0.027), and Esp-AF647 labeled proteins. The biofilms were washed with 500 μL TSB, incubated with 250 μL Syto 13 (1 μM) at 37°C for 15 min in the dark, and washed twice with PBS. Slides were prepared by removal of the chambers, followed by addition of 1 drop of ProlongGold antifade mounting solution and a coverslip. Slides were cured for 2 h at RT in the dark.

Biofilms were imaged using a Leica TCS-SPE confocal system with coded DMI4000B-CS inverted microscope (Leica, Wetzlar, Germany) using a 10x/0.30 NA HC PL Fluotar dry objective. Confocal images were obtained from each sample in two independent experiments with an average of 35 Z slices scanned with a 1.2 μm step size at a resolution of 512 x 512 pixels. Simultaneous dual-channel imaging was used to display Syto 13 (green) and AF647 (far red) fluorescence. Emission wavelengths for green fluorescence ranged between 495–553 nm and for red fluorescence between 650–670 nm. The excitation wavelength and laser power were 488 nm and 55% for the FITC (green) emission filter and 635 nm and 30% for the Cy5 (red) emission filter. The pinhole aperture was 488 nm-128.6 μm, 1.36 AU (frame average 1) and 635 nm-94.3 um 0.99 AU (frame average 4). The PMT detector gains and offsets used were 670 and -5 for the 488 nm laser line and 600 and -5 for the 635 laser line. Z-stack images were taken for each sample; the upper and lower stacks were set by cycling through the axial range, in both directions, until no fluorescence was observed (or the image was out of focus). LAS AF software was used for image acquisition and to export the data into individual image files (JPEG/TIFF). The Manders coefficient [25], which reports the ratio of the summed intensities from pixels in the red channel (protein) for which the intensity in the green channel (biofilm) was above zero to the total intensity in the red channel, was calculated for the entire stack with the ImageJ plug-in JACoP [26].

### SEC-MALS-SAX

Size exclusion chromatography (SEC) coupled multi-angle light scattering (MALS) and small-angle x-ray scattering (SAXS) was carried out at the SIBYLS beamline 12.3.1 at the Advanced

Light Source [27]. Esp fragments (80 μL, S4 Table) were loaded onto a Shodex KW803 SEC column, equilibrated in 100 mM NaCl, 100 mM Tris, pH 7.2 with a flow rate of 0.5 mL/min at 20˚C. SEC eluant was split 3:1 between a SAXS flow cell and MALS cell. Three second SAXS exposures were collected continuously at a wavelength of 1.127 Å, and with a sample-to-detector distance of 2.1 m. Buffer after the peak was averaged and used for subtraction. SAXS frames with consistent Rg were merged for further analysis. MALS was measured on an 18-angle DAWN HELEOS II light scattering detector connected in tandem to an Optilab refractive index concentration detector (Wyatt Technology), and was calibrated with BSA (45 μL, 10 mg/mL) in the same run. Data were processed with ASTRA Version 6.1.6.5 (Wyatt Technology) with d$n$/d$c$ set at 0.19. SEC-SAXS data were analyzed on SCÅTTER (https://bl1231.als.lbl.gov/scatter/); GNOM (ATSAS package) [28], and FoXS/FoXSDock [29]. *Ab initio* shape reconstructions were calculated using GASBOR in the ATSAS suite [30]. ColabFold models [31] were conformationally modified in ALLOSMOD-FOXS [32, 33] to fit the experimental data.

## Esp452 antibodies

Purified Esp452 was used as an antigen to generate rabbit polyclonal antibodies commercially (Cocalico Biologicals). The rabbit was boosted three times on days 14, 21, and 49 after initial inoculation, and the bleed collected on day 56 was used for experiments.

## Flow cytometry

Biofilms were dissolved with 1.5 M NaCl and *E. faecalis* pelleted from solution by centrifugation, as described above. Bacterial pellets were resuspended to an $OD_{600}$ of 1.0. Three hundred μL of the resuspension was blocked with PBS supplemented with 1% BSA (PBS+BSA) on ice for 30 min. Rabbit anti-Esp polyclonal antibody was added at a 1:500 dilution, and the sample was incubated for 1 h on ice. The sample was centrifuged (4,500 x $g$, 5 min, 4˚C), and the pellet was washed 3X with PBS. The pellet was resuspended in PBS+BSA with 1:200 donkey anti-rabbit IgG conjugated to Alexa Fluor 488 (Invitrogen), and incubated 30 min at RT. The sample was centrifuged (4,500 x $g$, 5 min, 4˚ C), and the pellet was washed with PBS. The sample was resuspended in 1 mL PBS+BSA and diluted 1:10 in PBS for analysis by flow cytometry (BD Accuri).

## Western blot

Samples were resolved on a 12% acrylamide SDS-PAGE gel and transferred to a nitrocellulose membrane (100 V, 45 min). The membrane was blocked in 0.1% TBST containing 1% w/v milk powder for 1 h at RT, then incubated in 0.1% TBST + milk containing 1:500 dilution of rabbit anti-Esp452 polyclonal antibodies at 4˚C overnight. The membrane was washed 3X with 0.1% TBST, then incubated in 0.1% TBST containing 1:4000 goat anti-rabbit antibody conjugated to HRP for 1 h at RT (SouthernBiotech).

## Enteropeptidase digestion of Esp743

Eight units of enteropeptidase light chain (New England Biolabs, 16 units per μL) and 25 μg of Esp743 (50 μg/μL) were added to 39 μL 50 mM NaCl, 2 mM $CaCl_2$, 20 mM Tris, pH 7.5 for 3 h at 37˚C. The reaction was stopped by incubation for 30 min at RT with 1.1 μL 200 mM PMSF, 1.1 μL 400 mM EGTA, and 2.8 μL 4 M NaCl, for final concentrations of 5 mM,10 mM, and 250 mM, respectively. SDS-PAGE loading dye was then added to samples, which were resolved by SDS-PAGE and visualized by InstantBlue staining.

## Mass spectrometry

Mass spectrometry measurements were conducted by the Molecular Mass Spectrometry Facility of the UC San Diego Chemistry and Biochemistry Department, using an Agilent 6230 LC-ESI-TOFMS or a Bruker Autoflex Max MALDI-TOFMS.

## Results

### Structure of Esp452

A portion of the N-terminal region of Esp encompassing aa 50–452, Esp452, was crystallized. This shorter portion of the N-terminal non-repeat region of Esp (Fig 1A) was chosen since potential additional repeat regions were identified downstream of aa 453 using RADAR [34]. Esp452 was recombinantly expressed in *E. coli* with its putative signal sequence retained at its N-terminus, as only a moderate probability of signal sequence cleavage (between 49 and 50) was predicted by SignalP [35]. Mass spectrometry (ESI) of Esp452, which was purified using a His$_6$-tag that had been added to its C-terminus, demonstrated that the predominant fraction started at aa 50 due to processing in *E. coli* (S1 Fig). The structure of Esp452 was determined to 1.4 Å resolution limit by single anomalous dispersion (SAD) from selenomethionine-labeled protein (S2 Table).

The structure revealed that Esp452 is composed of two globular domains (Esp1, aa 58–236; and Esp2, aa 257–447), each having a DE-variant immunoglobulin (DEv-Ig)-fold, in which two additional β-strands (D' and D") occur within the Ig-fold (Figs 1B, S2A and S2B). The two domains are similar but not identical (rmsd 3.3 Å, 183 Cα). Esp1 and Esp2 pack together, with a number of waters at the interface, and form a continuous surface, which is positively charged on one side and negatively charged on the other (S2C Fig). A 20-amino acid long loop, which appears to be flexible due to its high relative B-factor, connects the two domains (S2D Fig).

A structural similarity search [36] carried out soon after the structure was determined showed that Esp452 resembled proteins belonging to two large families of bacterial adhesins, namely the Microbial Surface Components Recognizing Adhesive Matrix Molecules (MSCRAMMs) and the *Streptococcus* Antigen I/II protein families (S2E and S2F Fig). MSCRAMMS have been identified in *Staphylococcus*, *Streptococcus*, and *Enterococcus*, and are typically involved in adhesion to host extracellular matrix proteins [37]. Eight MSCRAMMs that are structurally similar to Esp452 bind fibrinogen (Fg) [37–41], and thus Fg-binding by Esp452 was probed by ELISA. While binding between Fg and *S. pyogenes* M1 protein was detected (as a positive control), no binding between Fg and Esp452 was evident (S3 Fig). Antigen I/II proteins are virulence factors that mediate glycan-dependent attachment of *Streptococcus* to mucins or tissues [42]. The glycan-binding sites of Antigen I/II proteins are variable, but are typically characterized by a stabilizing isopeptide bond and a BAR motif. While Esp452 possessed neither of these structural features, it remained possible that Esp452 targeted glycans. Glycan-binding of Esp452 was tested using a human glycan microarray screen, but Esp452 did not bind glycans in the screen at a significant level or dose-dependent manner (S3 Table); these included sialic acid, lactose, and GalNAc, each of which bind Antigen I/II proteins [42, 43].

*E. faecalis* Esp is ascribed to have a role in urinary tract infections. This is based on comparison of parental MMH594 and MMH594b (Δ*esp*) strains in a murine model of infection [44]. This study reported that MMH594 was cleared less well in the urine and bladder compared to MMH594b (Δ*esp*), although no difference in histopathology was evident. Thus, with the possibility that a potential host target was in the urinary tract, we sought to verify this result. However, no difference was seen in bacterial counts between MMH594 and MMH594b (Δ*esp*) in

urine, bladder, or kidney (S4 Fig). Additionally, there were no differences between rates of bacterial presence or absence across tissues or time points. Together, these results indicated that Esp was unlikely to be an adhesin that targeted the urinary tract.

### Esp452 strengthens enterococcal biofilms

As Esp is implicated in biofilm production, the effect of Esp452 on enterococcal biofilms was evaluated. Log-phase cultures of MMH594b (Δ*esp*) were inoculated into media containing 0.5% glucose, as no effect of Esp is seen without glucose [5, 7, 8, 12, 13], and the resulting culture was placed into polystyrene wells. Esp452 or a fragment corresponding to the remaining portion of the N-terminal region, aa 453–743 (Fig 1A, $Esp_{453-743}$), was added to the wells at the time of inoculation. As a negative control, PBS was added instead of protein. After 19–20 h of growth to stationary phase, biofilms were evident in all wells (Fig 2A). Culture supernatants, which contained planktonic bacteria, were removed from biofilms, and the biofilms were washed twice. Biofilms that had Esp452 added were more resistant to washing than those that had either PBS or $Esp_{453-743}$ added (Fig 2A). Biofilms were then solubilized in 1.5 M NaCl [45], and bacteria were harvested from this biofilm fraction. Bacteria in the biofilm fraction along with those in the planktonic and two wash fractions were enumerated based on ATP content, as determined by luciferase-generated luminescence (Fig 2B). This measure was verified to be linearly related to the number of colony forming units and optical density (S5A–S5C Fig). Bacterial counts were enumerated rather than the more conventional approach of evaluating biofilms by CV staining, as the latter is non-specific and would detect protein added to biofilms, as in our experiments. The total luminescence of biofilms grown in the presence of PBS, Esp452, or $Esp_{453-743}$ was equivalent (S6 Fig), indicating that the addition of the Esp fragments had no effect on bacterial numbers.

While no difference in luminescence for planktonic fractions was observed, a statistically significant difference was observed for the two wash fractions, in which fewer bacteria were washed away from the biofilm grown with Esp452 as compared to the biofilms grown with $Esp_{453-743}$ or PBS (Fig 2B). Consistent with this result, significantly more bacteria were recovered from the biofilm grown with Esp452 as compared to those grown with $Esp_{453-743}$ or PBS (Fig 2B). These results indicated that Esp452 strengthened biofilms against mechanical disruption (i.e., washing).

The effect of Esp452 on biofilm strengthening was dose-dependent. The results described above were carried out with 4.0 μM Esp452 or $Esp_{453-743}$. We found that 4.0 μM Esp452 was saturating and that the $EC_{50}$ was approximately 740 nM (Fig 2C). In comparison, the addition of $Esp_{453-743}$ at the highest concentration of 8 μM continued to have no effect on biofilms as compared to the addition of PBS. For experiments described below, unless stated otherwise, Esp452 and other Esp fragments were used at the saturating concentration of 4.0 μM.

We next asked if Esp452 possessed the capacity to strengthen biofilms against other types of perturbations, such as enzymatic degradation by DNase. Enterococcal biofilms contain extracellular DNA and can be degraded by DNase [46, 47], and we confirmed that enterococcal biofilms were weakened by DNase, with the bacterial count decreased to 57.8% of an untreated biofilm (Fig 2D). By comparison, the bacterial count in biofilms grown with Esp452 and treated with DNase were reduced to only 77.0% (Fig 2D). Biofilms grown with $Esp_{453-743}$ were not significantly different from biofilms grown with PBS. These data indicated that Esp452 strengthened enterococcal biofilms from disruption caused by DNase.

Biofilm formation occurs in stages, beginning with attachment and initiation of biofilm matrix production [10]. We wondered if the strengthening provided by Esp452 was contingent upon its presence during the attachment and initiation phases. To test this, MMH594b (Δ*esp*)

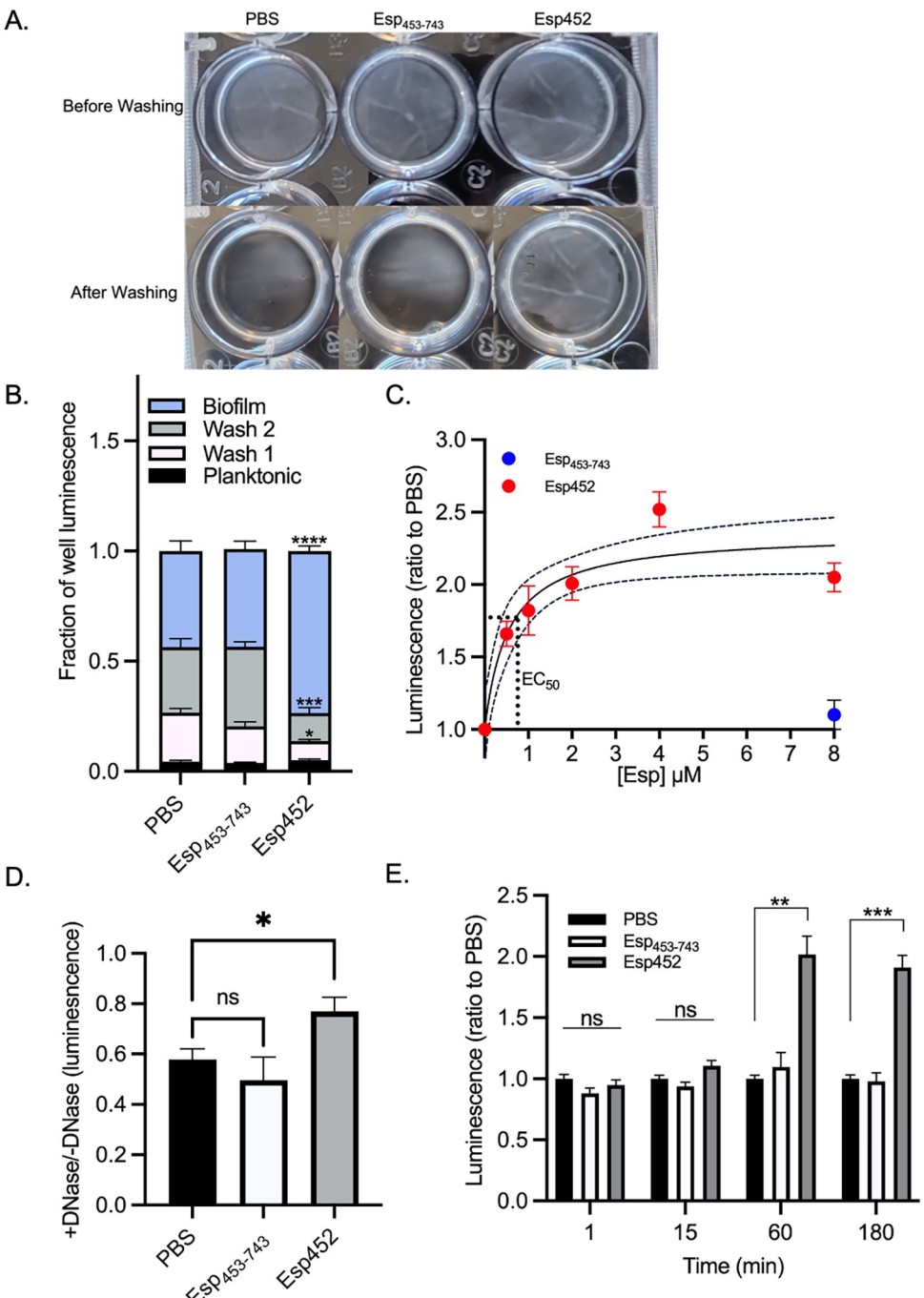

**Fig 2. Strengthening of enterococcal biofilms by Esp452. A.** MMH594b ($\Delta esp$) biofilms grown with PBS, Esp$_{453-743}$, or Esp452 before (top) and after (bottom) washing. **B.** Proportion of total luminescence from luciferase generated by the planktonic fraction, wash fractions, or biofilm fraction of *E. faecalis* MMH594b biofilms grown with PBS, Esp$_{453-743}$, or Esp452. Samples were collected in triplicate for each of three independent experiments. Each luminescence measurement was divided by the total luminescence of the corresponding well. The standard error of the mean (SEM) is indicated with error bars. Esp$_{453-743}$ and Esp452 fractions were compared to the corresponding PBS fraction by 2-Way ANOVA and Tukey's post hoc test. All significant results are indicated on the graph. $^{*}$ $p < 0.01$, $^{***}$ $p < 0.0001$, $^{****}$ $p < 0.00001$. **C.** Luminescence of biofilm fraction for MMH594b ($\Delta esp$) biofilms grown with Esp452 (red circles) at varying concentrations or with Esp$_{453-743}$ (blue circle) at 8 μM, divided by the average luminescence of biofilm fractions for biofilms grown with PBS. Samples were collected in duplicate for each of three independent experiments. The SEM is indicated with error bars. The 95% confidence interval (CI) of the dose-response curve is indicated in dashed lines. The EC$_{50}$ is indicated with the dotted line. **D.** Luminescence of biofilm extract of MMH594b biofilms

grown with PBS, $Esp_{453-743}$, or Esp452, and then incubated with DNase II for 3 hours before washing and dissolution, divided by the average luminescence of biofilm fractions of biofilms grown with water instead of DNase. The experiment was conducted with duplicates or triplicates in at least three independent experiments. The SEM is indicated with error bars. Samples were compared by Welch's ANOVA with Dunnett's T3 post-hoc test. $^*$ $p < 0.05$. **E.** Luminescence of the biofilm fraction from pre-grown MMH594b (Δ*esp*) biofilms to which PBS (black), $Esp_{453-743}$ (white), or Esp452 (gray) was added for the indicated time before washing and dissolution. Samples were collected in duplicate or triplicate for each of three independent experiments. Values were normalized at each time point to the luminescence of the PBS sample. The SEM is shown with error bars. The samples were compared by Welch's ANOVA with Dunnett's T3 post hoc test. $^{**}$ $p < 0.001$, $^{***}$ $p < 0.0001$.

biofilms were first grown for 19–20 h without the addition of Esp fragments, and then Esp452, $Esp_{453-743}$, or PBS was added to the pre-formed biofilms. Esp452 had a strengthening effect on pre-formed biofilms, which was evident after 60 min of incubation and required more than 15 minutes (Fig 2E). As before, no strengthening occurred with $Esp_{453-743}$ or PBS. These results indicated that Esp452 was not required during biofilm attachment or initiation for its strengthening action, and that Esp452 could strengthen pre-existing biofilms.

## Esp452 requires low pH for biofilm strengthening

The effect of glucose on biofilm strengthening by Esp452 was investigated. We found that reduction of the glucose concentration in the biofilm growth media from 0.50% to 0.18% had a dramatic effect. At 0.18% glucose, Esp452 failed to strengthen the biofilm (Fig 3A). *Enterococcus* is known to acidify growth media containing glucose through fermentation [48], and in line with this, the pH of the media overlying biofilms grown in 0.50% glucose was pH 4.56 ± 0.07, and pH 5.60 ± 0.06 for those grown in 0.18% glucose (Fig 3B). Esp truncation fragments had no effect on the pH of the overlying media (Fig 3B). These results raised the possibility that pH was a determinative factor in biofilm strengthening by Esp452.

To assess this possibility, Esp452 was placed in buffer solutions of varying pH. At pH 4.5 and lower, Esp452 aggregated with visible precipitation as monitored by $OD_{400}$, while $Esp_{453-743}$ did not aggregate even at the lowest pH tested, 4.2 (Fig 3C). Aggregation of Esp and the sequence-related *S. aureus* protein Bap is associated with the formation of β-rich amyloid-like structures [13, 49]. Thioflavin T (ThT), a fluorescent dye that exhibits an emission shift when bound to β-rich protein aggregates, was used to evaluate the formation of amyloid-like structures by Esp. Esp452 bound ThT at pH 4.5 and lower (Fig 3D), while $Esp_{453-743}$ bound little ThT between pH 4.2–6.0. These results suggested that the acidic pH brought about by fermentation of 0.5% glucose was crucial for biofilm strengthening by Esp452, and triggered the unfolding of Esp452 to form aggregates and precipitates, including amyloid-like structures.

## Esp452 biofilm strengthening inhibited within intact N-terminal region

The entire N-terminal region of Esp (Fig 1A), Esp743, was recombinantly expressed and purified, and assayed for biofilm strengthening. Surprisingly, even though Esp743 contained Esp452, Esp743 provided no biofilm strengthening (Fig 4A). In line with this result, Esp743 showed no aggregation or binding to ThT at pH 4.5 (Fig 3C and 3D). In addition, Esp452 added to biofilms (grown with 0.5% glucose) bound ThT in a dose-dependent manner within these biofilms, whereas neither Esp743 nor $Esp_{453-743}$ did (Fig 4B). Fluorescence microscopy of fluorophore-labeled Esp fragments, which were added at the time of inoculation, revealed that Esp452, but not Esp743, colocalized with biofilms, which were detected using a nucleic acid-sensitive fluorophore (Fig 4C and S1–S3 Files). Colocalization was confirmed quantitatively in two independent experiments, for which the Manders coefficient [25] was calculated to be 0.44 and 0.47 for Esp452, but only 0.03 in both experiments for Esp743. Esp452 was present at and

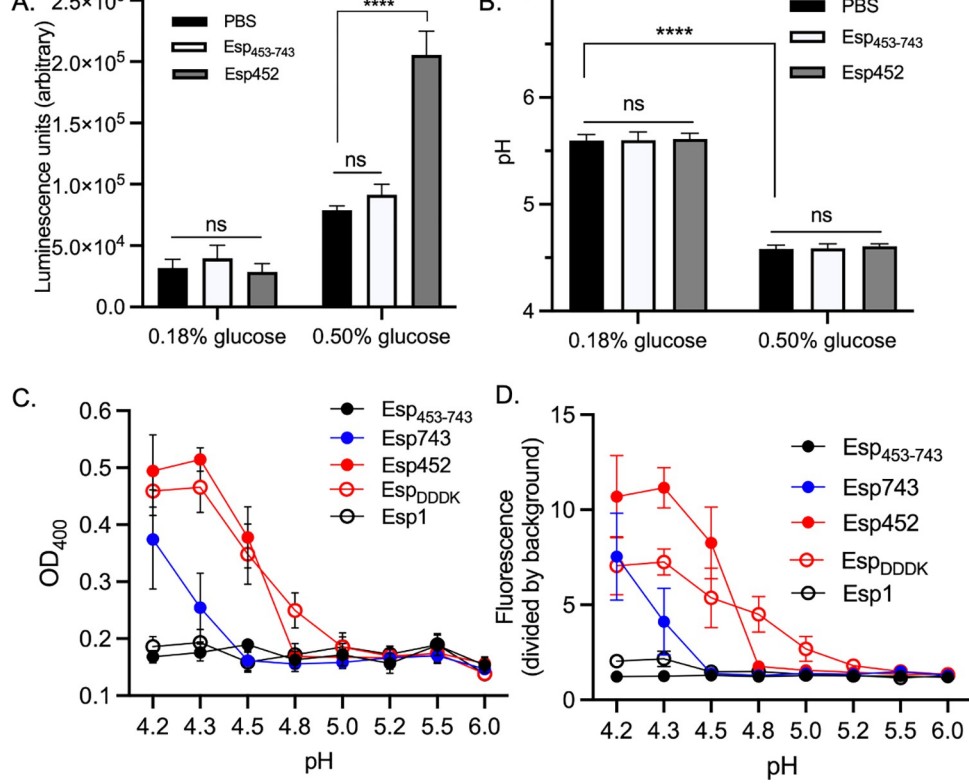

**Fig 3. Low pH promotes biofilm strengthening by Esp452. A.** Luminescence of MMH594b (Δ*esp*) biofilms grown in either 0.18% or 0.50% glucose, and with PBS, Esp$_{453-743}$, or Esp452. The experiment was conducted with triplicates. Samples from three independent experiments were compared by 2-way ANOVA and Tukey's test. **** $p < 0.0001$. **B.** The pH of biofilm cultures of MMH594b grown in TSB with 0.18 or 0.50% glucose, and with Esp452, Esp$_{453-743}$, or PBS. The experiment was conducted with triplicates. Samples from three independent experiments were combined and compared by 2-way ANOVA and Tukey's test. **** $p < 0.0001$. **C.** OD$_{400}$ of Esp fragments (25 μg) incubated in sodium citrate buffer at the indicated pH for 24 h. Experiments were conducted with duplicates. Samples from three independent experiments were combined and the SEM is indicated with error bars. **D.** Fluorescence of Thioflavin T incubated for 24 h with Esp fragments (25 μg) in sodium citrate buffer at the indicated pH. Experiments were conducted with duplicates. Samples from three independent experiments were combined and the SEM is indicated with error bars.

near the top of the biofilm, which may have resulted from precipitation from solution when the biofilm reached a sufficiently low pH. Together, these results suggested that in the context of Esp743, the region corresponding to Esp$_{453-743}$ inhibited the aggregation, amyloid-like structure formation, and biofilm strengthening activities of Esp452.

## SEC-MALS-SAXS analysis

To investigate Esp452 within the context of the entire N-terminal region, crystallization of Esp743 was pursued. While crystals were obtained, no X-ray diffraction was observed and efforts to improve the quality of the crystals were unsuccessful. Thus, size-exclusion chromatography (SEC) coupled to multi-angle light scattering (MALS) and small-angle x-ray scattering (SAXS) analysis was carried out for Esp743, along with Esp452 and Esp$_{453-743}$. At pH 7.2, all three Esp fragments eluted primarily as single, monodisperse peaks, and data were linear on a Guinier plot (S7 Fig), indicative of a lack of aggregation. The MALS and SAX molecular masses of these fragments were consistent with monomeric states (S4 Table). The Porod exponent (P$_X$), a protein density measure (maximum of 4.0 corresponds to a well-folded protein, and

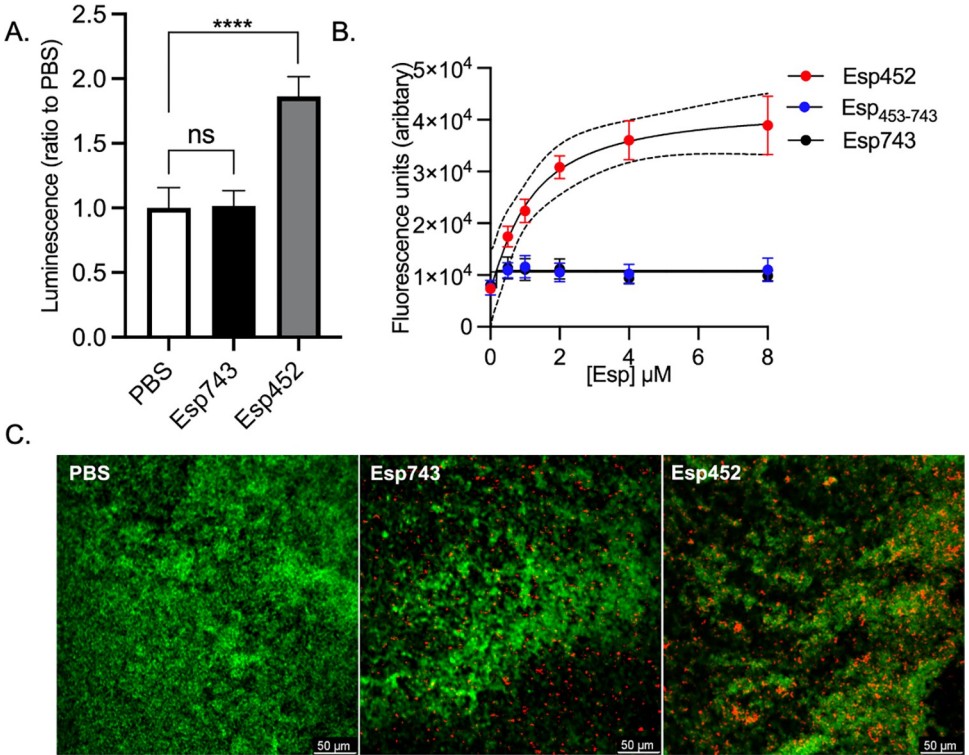

**Fig 4. Esp452 inhibited within intact N-terminal region. A.** Luminescence of biofilm fractions of MMH594b (Δ*esp*) biofilms grown with PBS, Esp743, or Esp452, divided by the average luminescence of biofilm fractions from biofilms grown with PBS. The experiment was conducted with triplicates in three independent experiments. The SEM is indicated with error bars. Samples were compared by Welch's ANOVA and D3 Dunnett's post hoc test. ****
$p < 0.0001$. **B.** Fluorescence of Thioflavin T added to MMH594b (Δ*esp*) biofilms that were grown with Esp452, Esp$_{453\text{-}743}$, or Esp743. Fluorescence measurements from three independent experiments were combined. The SEM is indicated with error bars. The 95% CI for the dose-response curve of Esp452 is shown as dashed lines. **C.** Confocal microscopy images of MMH594b (Δ*esp*) biofilms grown with PBS, Esp743-AF647, or Esp452-AF647 (red). Biofilms were stained with Syto-13 (green). The images represent Z-stacks with the greatest number of red pixels.

value of 2.0 corresponds to an unfolded protein), indicated that Esp452 and Esp743 were both well-folded ($P_X = 4.0$ and 3.8, respectively), whereas Esp$_{453\text{-}743}$ showed significant flexibility ($P_X = 2.5$) (S4 Table) [50].

SAXS data for Esp452 in solution did not match that calculated from the crystal structure (Fig 5A and 5B). In particular, the solution data were missing maxima and minima that were present in the calculated crystal structure curve, indicating an oblong shape for Esp452 in solution as opposed to the rounded globular structure observed in the crystal structure. Rearrangement of Esp1 and Esp2 domains into alternative configurations, as identified with FoXSDock, resulted in an improvement in the $\chi^2$ fit to the data from 110 to 3.5–4 (Fig 5C). The flexibility of the loop connecting Esp1 and Esp2 is consistent with this rearrangement (S2D Fig). SAXS data for Esp$_{453\text{-}743}$ was consistent with an extended, multidomain structure. *In silico* prediction using ColabFold [31] suggested that Esp$_{453\text{-}743}$ is composed of three domains, an Ig-like domain followed by two Rib domains [51]. The fit of this *in silico* model to the data was improved from a $\chi^2$ of 4.4 to 1.8 through molecular dynamics and an ensemble of two models (Fig 5C, models A and B), consistent with the flexibility of Esp$_{453\text{-}743}$. For Esp743, *ab initio* shape reconstruction using SAXS data indicated a globular head connected to an extended tail [30]. Most significantly, the Esp743 reconstruction was shorter than the sum of the Esp452 and Esp$_{453\text{-}743}$ reconstructions. Together, these data suggested that the Esp452 and Esp$_{453\text{-}743}$

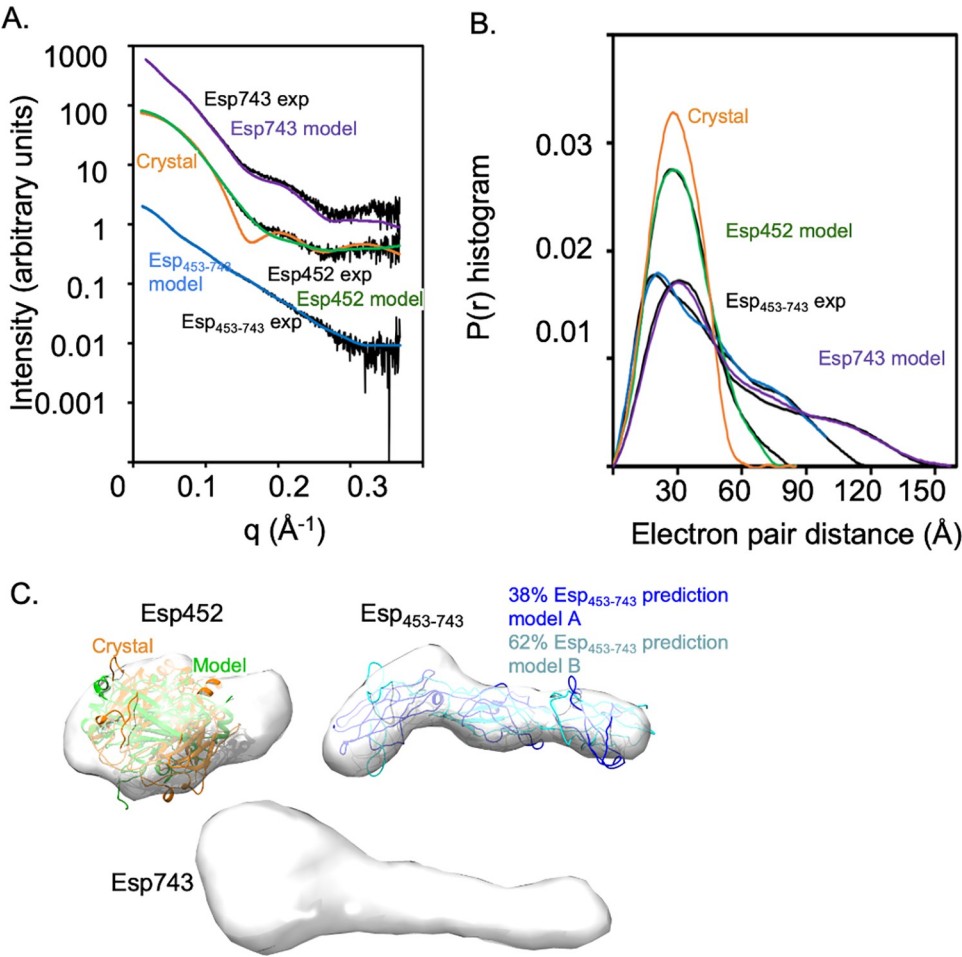

**Fig 5. SAXS of Esp fragments. A.** Experimental SAXS data (exp) in reciprocal space for Esp452, Esp453-743, and Esp743 overlaid with SAXS curves calculated for the crystal structure of Esp452 (orange) or models of Esp452, Esp453-743, or Esp743 (green, blue, purple, respectively). **B.** Same as A, except in real space and only experimental data and SAXS curves predicted from models are shown. Curves were normalized for area under the curve. **C.** Shape reconstructions based on SAXS data of Esp452, Esp453-743, and Esp743 overlaid with Esp452 or Esp453-743 models.

portions interacted in the context of Esp743. This interaction provided a likely mechanism for stabilization of the Esp452 domains within Esp743, resulting in the inability of the Esp452 domains when a part of Esp743 to unfold, form amyloid-like structures, and strengthen biofilms at pH 4.5.

## Release of biofilm strengthening fragments

Release of Esp452 from intact Esp would provide one mechanism of biofilm strengthening. A number of proteases, such as several matrix metalloproteases, chymotrypsin, and granzyme B, are predicted to cleave Esp743 within aa 445–455, based on sequence analysis [52]. Intriguingly, Esp contains the highly specific enteropeptidase cleavage sequence DDDK (aa 223–226) within Esp1 on a loop connecting its last two β-strands (S8 Fig). This sequence is predicted based on structure and sequence analysis [52] to be amenable to cleavage, which would result in a fragment smaller than Esp452. To assess whether the DDDK site was accessible to enteropeptidase, Esp743 was treated with human enteropeptidase light chain, and a fragment consistent with the cleavage at this site was produced (S9A and S9B Fig). To explore the activity of

this specific fragment, an Esp fragment corresponding to the enteropeptidase cleavage product, called Esp$_{DDDK}$ (Fig 1A), was recombinantly expressed and purified. This fragment was processed at its N-terminus in *E. coli* such that it spanned aa 50–226 (S10A Fig). When added to enterococcal biofilms, Esp$_{DDDK}$ strengthened biofilms to nearly a similar extent as Esp452 (Fig 6A). Notably, Esp$_{DDDK}$ aggregated and bound ThT at pH 4.8–5.0 (Fig 3C and 3D), about half a unit higher than Esp452, suggesting that Esp$_{DDDK}$ was more unstable than Esp452.

We hypothesized that the relative instability of Esp$_{DDDK}$ may have been due to the fact that this fragment was missing a part of the Esp1 domain. To test this, the entire Esp1 domain was recombinantly expressed and purified; this fragment was processed at its N-terminus in *E. coli* such that it spanned aa 50–241(S10B Fig). Consistent with this hypothesis, Esp1 did not aggregate and bound very little ThT even at the lowest pH tested, 4.2 (Fig 3C and 3D), and provided no biofilm strengthening (Fig 6B).

These results suggested that the stability of Esp fragments, based on the precise site of proteolytic cleavage, determined the ability and pH threshold of such fragments to strengthen biofilms. The key role of protein stability in biofilm strengthening was consistent with the formation of amyloid-like structures, which require protein unfolding.

## Esp743 strengthens biofilms at pH $\leq$ 4.3

The experiments presented above revealed that Esp743 was capable of aggregating and binding ThT at pH $\leq$ 4.3 (Fig 3C and 3D). This suggested that Esp743 should strengthen biofilms in an *Enterococcus* strain that acidified the medium to a lower pH than MMH594. The *E. faecalis* strains FA2-2 and OG1RF, neither of which encode *esp*, were grown in the presence of 0.5% glucose, and both were confirmed to form biofilms. The pH of media overlying the biofilms was determined. FA2-2 acidified the media to pH 4.08 ± 0.16 and OG1RF to 4.27 ± 0.02. In agreement with the predictions above, Esp743 had a statistically significant effect on strengthening biofilms formed by FA2-2 (Fig 6C) and OG1RF (Fig 6D). As expected, Esp452 also strengthened FA2-2 and OG1RF biofilms whereas Esp$_{453-743}$ had no effect. A further prediction was that there should be no difference in biofilm strength between MMH594 and MMH594b (Δ*esp*). This was because the pH threshold for Esp743 was $\leq$ 4.3 and MMH594 had been found to acidify the media to pH ~4.5. Additionally, no N-terminal fragments of Esp that might have a higher pH threshold were produced in the biofilm (S11 Fig). Consistent with these observations, there was no difference in biofilm strength between MMH594 and MMH594b (Δ*esp*) (Fig 6E). Surface expression of Esp in MMH594 was verified by FACS (S12A Fig).

As noted above, plasmid-borne expression of *esp* from its native promoter has been shown to favor biofilm production by FA2-2 [8]. This result was verified in FA2-2 using the same plasmid, pEsp, through both CV staining and bacterial counts (S13 Fig). However, we found that Esp was expressed on the surface of the transformed FA2-2 strain to a level ~10-fold greater than native expression on the surface of MMH594 (S12B Fig). While the pH of the transformed FA2-2 strain was the same as untransformed FA2-2, and no N-terminal fragments of Esp were detected in biofilms of transformed FA2-2 (S14 Fig), overexpression raised the concern that biofilm strengthening in FA2-2 may be artifactual. This conclusion was verified by transforming pEsp into MMH594. Esp was overexpressed beyond native levels in transformed MMH594 (S12B Fig), and transformed MMH594 produced biofilms that were significantly strengthened as compared to untransformed MMH594 and MMH594b (Δ*esp*) (S15 Fig). Thus, native expression levels of Esp did not strengthen biofilms but overexpressed levels did, indicating that overexpression of Esp leads to artifactual strengthening of biofilms.

Lastly, we asked whether MMH594 would acidify the media further if the glucose concentration were increased. Indeed, the pH of the media overlying an MMH594 biofilm grown in

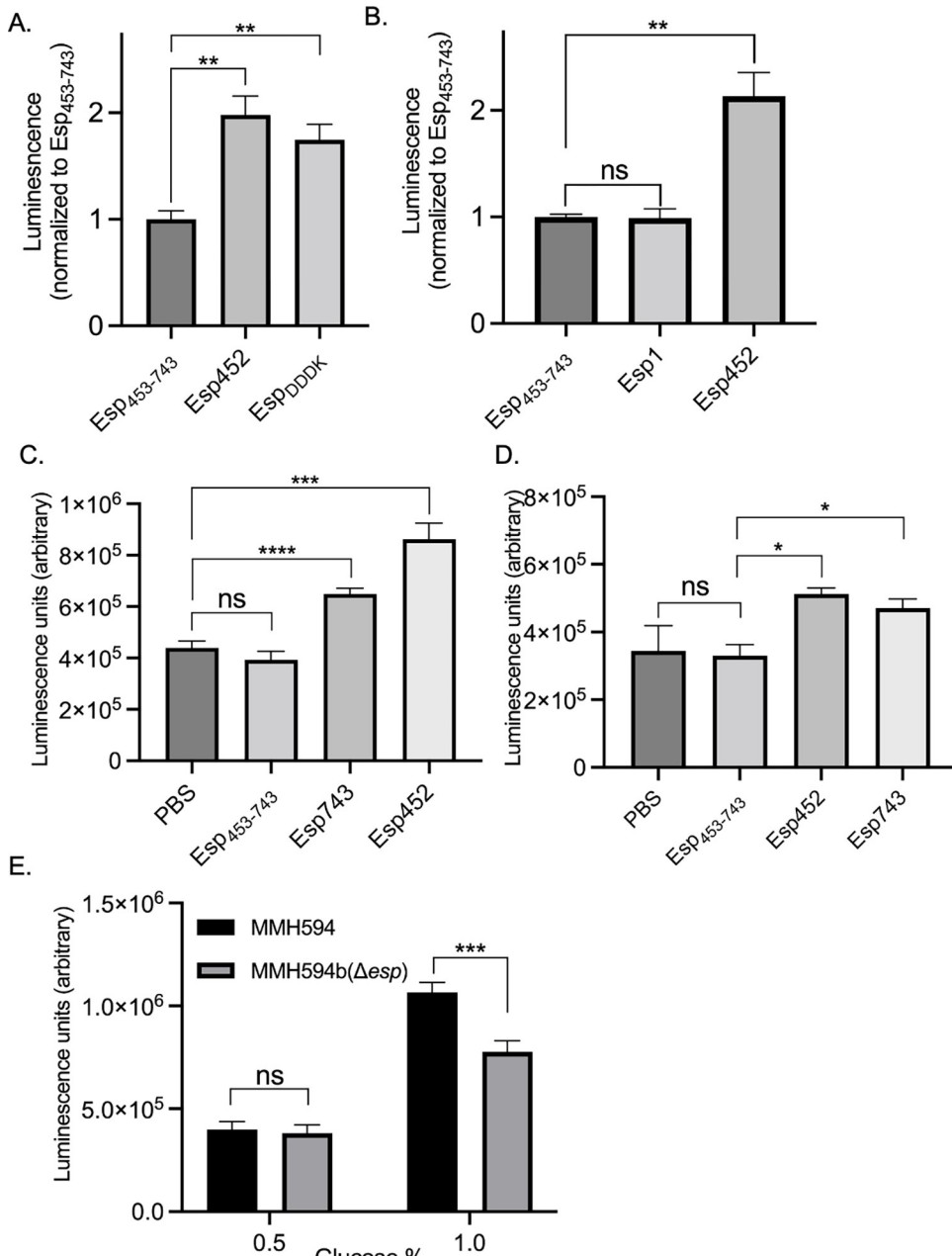

**Fig 6. Biofilm Strengthening by Esp_DDDK and Esp743. A.** Luminescence of MMH594b (Δ*esp*) biofilms grown with
Esp453-743, Esp452, or Esp_DDDK. Luminescence values were divided by the average luminescence of biofilms grown
with Esp453-743. The experiment was performed in triplicate in three independent experiments. Samples were
compared by Welch's ANOVA with D3 Dunnett's post-hoc test. **, p < 0.01. **B.** Luminescence of MMH594b (Δ*esp*)
biofilms grown with Esp453-743, Esp452, or Esp1. All luminescence values were divided by the average luminescence of
biofilms grown with Esp453-743. The experiment was performed in duplicate or triplicate in three independent
experiments. Samples were compared by Welch's ANOVA with D3 Dunnett's post-hoc test. **, p < 0.01. **C.**
Luminescence of FA2-2 biofilms grown overnight with PBS, Esp453-743, Esp743, or Esp452. The experiment was
conducted with duplicates or triplicates in two or more independent experiments, with n ranging from 5 to 15. The
SEM is indicated with error bars. Samples were compared by Welch's ANOVA and D3 Dunnet's post hoc test.
p < 0.001, ***; p < 0.0001, ****. **D.** Luminescence of OG1RF biofilms grown with PBS, Esp453-743, Esp452, or Esp743.
The experiment was conducted with triplicates. Samples were compared by Welch's ANOVA with D3 Dunnett's post-
hoc test. * *p* < 0.05. **E.** Luminescence of MMH594 and MMH594b (Δ*esp*) biofilms grown in media supplemented with
either 0.5 or 1.0% glucose. The experiment was performed in triplicate in three independent experiments. Samples
were compared by 2-Way ANOVA with Tukey's post-hoc test. ***, p < 0.001.

1% glucose was 4.21 ± 0.03 rather than ~4.6 for one grown in 0.5% glucose. The increase in glucose concentration from 0.5% to 1.0% resulted in a slight increase in bacterial numbers, but there was no difference between MMH594 and MMH594b (Δ*esp*) at either concentration (S16 Fig). Consistent with the unfolding of Esp743 at pH ≤ 4.3, MMH594 biofilms were significantly strengthened compared to MMH594b (Δ*esp*) biofilms at 1% glucose (Fig 6E). These results provided evidence that the properties observed for the soluble Esp743 fragment were recapitulated by intact Esp attached to the enterococcal cell wall.

## Discussion

The role of Esp in enterococcal disease has been a subject of debate over the last several decades. Esp is encoded on a large pathogenicity island (~150 kilobases) in *E. faecalis* and *faecium* [2–5], and is found more frequently in clinical as compared to commensal strains [53, 54]. A role for Esp in biofilm production was originally suggested due to its sequence similarity to *S. aureus* Bap, a biofilm-forming protein [7]. Experimental evidence was garnered to support a role for Esp in biofilm production [7]. A deletion of *esp* in the clinical *E. faecium* E1162 strain led to decreased biofilm production, as assayed by CV staining, and this loss of function was complemented by plasmid-borne expression of *esp* from a heterologous, constitutive promoter [5]. Deletion of *esp* in certain *E. faecalis* strains led to decreased biofilm production, although no complementation was carried out in these cases [7]. However, other results raised doubts about the role of Esp in biofilm production. Notably, deletion of *esp* in several *E. faecalis* strains had little or no effect on biofilm production [7], several biofilm-forming *E. faecalis* strains naturally lacked *esp* [16], and a number of *E. faecalis* isolates that carried *esp* did not form biofilms [7, 17, 18].

To understand the function of Esp, we took a biochemical and structural approach. The N-terminal non-repeat region of Esp (i.e., Esp743) was identified by SAXS to be composed of a globular head connected to an extended tail. The globular head corresponded to two DEv-Ig domains, Esp1 and Esp2, which together constituted Esp452 and whose structure was determined to atomic resolution by X-ray crystallography. These two domains were followed in Esp743 by three domains, an Ig-like and two Rib domains (which together formed $Esp_{453\text{-}743}$), as predicted by *in silico* means. Significantly, the SAXS shape reconstruction of Esp743 was shorter than the sum of its two component parts, Esp452 and $Esp_{453\text{-}743}$. This provided evidence that the Ig-like domain (at the N-terminus of the $Esp_{453\text{-}743}$ portion) interacted with the globular head (i.e., Esp452).

DEv-Ig domains are common among bacterial adhesins, and while it remains possible that these domains in Esp have host targets, we instead identified a function that required unfolding of these domains into amyloid-like structures that strengthen biofilms. The structurally related proteins Bap and Antigen I/II also have DEv-Ig folds and form amyloid-like structures that localize to biofilms [49, 55–57], indicating that DEv-Ig folds can have biofilm-related functions in addition to adhesive ones. We found that Esp743 strengthened biofilms, as did Esp452 and the shorter $Esp_{DDDK}$ fragment, suggesting that this capacity is resident at the very N-terminal portion of Esp. Importantly, our biochemical approach enabled the addition of Esp fragments prior to and following biofilm growth. Based on this, we found that Esp was not required during biofilm initiation or growth, but instead provided strengthening, even to mature biofilms.

Esp452 colocalized with the enterococcal biofilm and bound ThT in this location, indicating an amyloid-like structure for this Esp fragment in the biofilm. For the various Esp fragments examined, biofilm strengthening correlated with protein aggregation, precipitation, and ThT-binding. $Esp_{DDDK}$ was the least stable with a pH threshold of ~5. $Esp_{DDDK}$ constitutes an

incomplete domain in that it lacks the last β-strand of the Esp1 DEv-Ig fold. By comparison, Esp1, the complete domain, did not aggregate or precipitate, and bound very little ThT at the lowest pH studied, 4.2. Esp452 had greater stability than $Esp_{DDDK}$ with a pH threshold of 4.5 and Esp743 even greater stability with a pH threshold of 4.3. The difference between Esp743 and Esp452 may be explained by stabilizing interactions between the extended tail in Esp743 with the globular head (i.e., Esp452), as demonstrated by SAXS. Intact Esp on the enterococcal surface had a low pH requirement, similar to that of Esp743, as MMH594 biofilms showed strengthening compared to MMH594 (Δ*esp*) biofilms at pH 4.2 but not 4.5 (grown in 1.0% vs. 0.5% glucose, respectively). The similar behaviors of Esp743 and intact Esp is consistent with Esp being a member of the "periscope" family [6], with C-terminal repeat domains serving a structural purpose of projection but not influencing the functional properties of the N-terminal region.

These results are consistent with those of Taglialegna et al. [13], who used an Esp fragment spanning aa 67–511. The N-terminus of this particular fragment lacks a few amino acids of the Esp1 domain which precede the first β-strand of the DEv-Ig fold, and the C-terminus is predicted to include only a portion of the Ig-like domain of $Esp_{453-743}$. However, it appears that this portion of the Ig-like domain suffices, as Esp 67–511 aggregates and binds ThT along with other amyloid-like indicators at pH ≤ 4.2 [13], suggesting it is equally as stable as Esp743. Esp 67–511, when expressed from a plasmid in *S. aureus*, results in biofilm production, as determined qualitatively through CV staining [13].

Similarly, intact Esp or Esp743, when expressed from a plasmid in *E. faecalis* FA2-2, has been reported to result in increased biofilm production, as quantitatively assayed by CV staining [12]. We verified this last result by using the same plasmid and finding that expression of intact Esp in FA2-2 resulted in strengthened biofilms based on bacterial counts. However, we also found by FACS that Esp was overexpressed by ~10-fold on the enterococcal surface by transformed FA2-2. A similar effect of *esp* overexpression was found for MMH594. While MMH594 and MMH594b (Δ*esp*) did not differ in biofilm strength (grown with 0.5% glucose), plasmid-borne overexpression of *esp* in MMH594 led to significant strengthening of the biofilm compared to MMH594b (Δ*esp*). Thus, overexpression had an artifactual effect on biofilm strengthening. The basis for overexpressed Esp strengthening biofilms is not known, but could be due to destabilization of Esp through molecular crowding on the surface. These results suggest that prior conclusions based on plasmid-borne expression of Esp in which surface expression was not monitored, along with experiments in which pH was not monitored should be taken with caution. At the same time, these results also suggest that natural variation in Esp expression level could account for strain-to-strain differences observed for Esp action.

The dependence on low pH for biofilm strengthening explains why glucose was necessary in the media to observe an effect of Esp in past reports. Fermentative metabolism of glucose results in the acidification of the extrabacterial environment, and low pH was necessary for biofilm strengthening by Esp fragments. It is conceivable that such low pH values would be encountered in the absence of glucose, as *Enterococcus* can metabolize a large variety of carbohydrates to produce lactic acid [58]. Such acidic conditions may be independent of bacterial fermentation, and instead encountered on abiotic surfaces or in the host. For example, the duodenum, where *Enterococcus* exists [59] and where enteropeptidase is found, can have a pH 4–5 following a meal [60]. Likewise, *Enterococcus* is a cause of dental disease, and dental lesions such as caries reach pH's as low as 4.4 [61]. The pH at which Esp exerts its biofilm strengthening activity may also be modulated by host proteases.

In summary, we show that Esp in acidic conditions provides significant strengthening to biofilms, resulting in retention of *Enterococcus* within perturbed biofilms. As biofilms have

properties that favor virulence, such retention is likely to favor the pathogenic potential of *Enterococcus*.

## Supporting information

**S1 Fig. Mass Spectrometry of Esp452.** ESI-TOF mass spectrum of recombinant Esp452-His$_6$ with the His$_6$-tag removed by PreScission protease digestion. The sequence and predicted mass of the construct is indicated on the right. The sequence LEVLFQ at the C-terminus is from the PreScission protease cleavage site. Esp sequence numbers are indicated.
(PDF)

**S2 Fig. Structure of Esp452.** A. Topology of Esp1 (top) and Esp2 (bottom) shown in rainbow coloring, with similar coloring of the domains shown in ribbon representation at right. B. Esp1 (pink) and Esp2 (blue) superposed and depicted as Cα traces. C. Molecular surface of Esp452 viewed perpendicularly to the β-sheets. Surfaces with negative character shown in red, neutral in white, and positive in blue, ranging from -4.0 to 4.0 kT. Shown below is the same view in ribbon representation. D. Z-score for Cα atom B-factors, calculated with the following formula: $Z = (B_x - B_{avg})/s$, where $B_x$ is the B-factor of a given Cα atom, $B_{avg}$ is the average B-factor of all Cα atoms in the structure, and $s$ is the standard deviation of the Cα B-factors. Esp1 is shown in pink, Esp2 in lavender, and the linker in black. Certain loops and helices connecting β-strands, as well as the N- and C-termini, also have higher than average B-factors. E. Superposition of Esp2 (blue) with DEv-Ig domains of ClfA (pink) and ClfB (cyan), which bind fibrinogen. Rmsd of 3.8 and 3.9 Å, respectively, with Esp2 for 183 Cα. F. Superposition of Esp1 (pink) with DEv-Ig domain (gray) of Antigen I/II. Rmsd of 3.5 Å with Esp1 for 182 Cα.
(PDF)

**S3 Fig. Fibrinogen binding.** Wells of an ELISA plate were coated with fibrinogen, and equivalent molar amounts of Esp452-His$_6$ or M1-His$_6$ protein, or a PBS control was added to the wells. Bound His$_6$-tagged proteins were quantified by ELISA using anti-His antibodies. The experiment was conducted one time in triplicate. Samples were compared by 1-Way ANOVA. *** $p < 0.001$.
(PDF)

**S4 Fig. Murine UTI model.** Mice were inoculated through the urethra with either MMH594 or MMH594b (Δ*esp*). At 1, 3, or 5 days after inoculation, urine was collected, mice were sacrificed and tissues were homogenized in PBS and plated. Data are shown as CFU/g of tissue or CFU/mL of urine. The experiment was performed with five mice per sacrifice day, and independent experiments were performed twice for 1 and 3 days and once for 5 days. The mean is indicated with a horizontal line. Samples were compared by Fisher's exact test. NS, $p > 0.05$.
(PDF)

**S5 Fig. Luminescence and bacterial counts.** (A) Luminescence and (B) CFU/mL as a function of OD$_{600}$, and (C) luminescence as a function of CFU/mL were measured for serial dilutions of MMH594. The Pearson correlation coefficient and corresponding $p$ value are indicated on each graph. These relationships also apply to MMH594 (Δ*esp*), as MMH594 and MMH594b (Δ*esp*) were confirmed to have the same growth kinetics, as reported previously [44].
(PDF)

**S6 Fig. Total luminescence of biofilm cultures.** The total luminescence values of each well in three independent experiments are shown. Samples in each experiment were compared by Welch's ANOVA with Dunnett T3 post hoc test. Total luminescence varied between experiments due to a variety of factors, including temperature at the time of measurement and age of

the luminescence reagent.
(PDF)

**S7 Fig. Guinier Plot.** Guinier plots for experimental SEC-SAXS data for Esp743, Esp452, and Esp$_{453-743}$.
(PDF)

**S8 Fig. DDDK.** The DDDK sequence (red) is located on a loop between the Esp1 F and G β-strands (blue and purple, respectively).
(PDF)

**S9 Fig. Enteropeptidase cleavage of Esp743. A.** Products of digestion of Esp743 at 37˚C for 3 h with the light chain of human enteropeptidase (EP) resolved by SDS-PAGE and InstantBlue-stained. At 3 h, the reaction was quenched before being applied to SDS-PAGE. The fragment designated by the arrowhead has a size matching Esp$_{DDDK}$. For the input sample, Esp743 and EP were incubated separately at 37˚C for 3 h, quench solution was added to each, and the two were added together and immediately applied to SDS-PAGE. **B.** MALDI-TOF mass spectrogram of enteropeptidase-digested Esp743 and the theoretical sequence and weight of a hypothetical enteropeptidase cleavage product. The peak corresponding to the theoretical mass is indicated on the spectrum with an arrow.
(PDF)

**S10 Fig. Mass Spectrometry of Esp$_{DDDK}$ and Esp1.** ESI-TOF spectra and sequences of (**A**) Esp$_{DDDK}$ and (**B**) Esp1. ESI-TOF of recombinant Esp-His$_6$ constructs in which the His$_6$-tag was removed by PreScission protease digestion. The sequences and predicted masses of the constructs are indicated on the right. The C-terminal sequence LEVLFQ is from the PreScission protease cleavage site. The peaks corresponding to Esp$_{DDDK}$ or Esp1 are indicated on the spectra with an arrow.
(PDF)

**S11 Fig. Western blots of MMH594 biofilms.** MMH594 and MMH594b (Δ*esp*) biofilms were grown with Esp452, Esp743, or PBS. The biofilms were dissolved with NaCl, filtered, and assayed for the presence of Esp by western blot using anti-Esp452 polyclonal antibodies.
(PDF)

**S12 Fig. FACS Analysis of Surface Expression of Esp. A.** MMH594 (left) and MMH594b (Δ*esp*) (right) isolated from biofilms were incubated with rabbit anti-Esp452 antibodies followed by secondary antibodies conjugated to Alexa Fluor 488 (orange). Bacteria with no antibodies (red) and with secondary antibody only (blue) were measured to assess background fluorescence. Data were graphed with FloJo. **B.** FA2-2 (orange), FA2-2 (pEsp) (green), MMH594 (red), and MMH594 (pEsp) (cyan) isolated from biofilms were incubated with rabbit anti-Esp452 antibodies followed by secondary antibodies conjugated to Alexa Fluor 488. Data were graphed with FloJo.
(PDF)

**S13 Fig. Effect of pEsp on FA2-2 biofilms.** Crystal violet (**A**) and luminescence (**B**) measurements of biofilms produced by *E. faecalis* FA2-2 with and without pEsp. The experiments were conducted with sextuplicates and triplicates, respectively. Samples were compared by Student's t-test. $p < 0.05$, $^*$; $p < .0001$, $^{****}$.
(PDF)

**S14 Fig. Western blot of FA2-2 biofilms.** FA2-2 and FA2-2 (pEsp) biofilms were grown with PBS or Esp452. The biofilms were dissolved with NaCl, filtered, and assayed for the presence

of Esp by western blot using anti-Esp452 polyclonal antibodies.
(PDF)

**S15 Fig. pEsp strengthens MMH594 biofilms.** Luminescence of biofilm fractions of MMH594b (Δ*esp*), MMH594, and MMH594 (pEsp). Welch's ANOVA and D3 Dunnet's post hoc test. p < 0.01, **.
(PDF)

**S16 Fig. Growth of MMH594 and MMH594b (Δ*esp*) in media containing 0.5 or 1.0% glucose.** Luminescence of entire biofilm cultures, including the planktonic fractions, of MMH594 and MMH594b (Δ*esp*) were measured. Samples were compared by 2-Way ANOVA and Tukey's posthoc test. p < 0.001, ***.
(PDF)

**S1 Table. Primers.**
(PDF)

**S2 Table. X-ray data collection and refinement statistics.**
(PDF)

**S3 Table. Glycan binding by Esp452.**
(PDF)

**S4 Table. SEC-MALS-SAX Analysis.**
(PDF)

**S1 File. Confocal images of biolfilms with PBS.** Individual Z-stacks of MMH594b (Δ*esp*) biofilms grown overnight with PBS. Bacteria were stained with Syto-13 (green) and proteins were labeled with AF647 (red). The first image was taken at the interface of the biofilm with the slide and each stack is 1.2 μm higher, progressing up to the top of the biofilm at the biofilm-media interface.
(PDF)

**S2 File. Confocal images of biofilms with Esp743-AF647.** Individual Z-stacks of MMH594b (Δ*esp*) biofilms grown overnight with Esp743-AF647. Bacteria were stained with Syto-13 (green) and proteins were labeled with AF647 (red). The first image was taken at the interface of the biofilm with the slide and each stack is 1.2 μm higher, progressing up to the top of the biofilm at the biofilm-media interface.
(PDF)

**S3 File. Confocal images of biofilms with Esp452-AF647.** Individual Z-stacks of MMH594b (Δ*esp*) biofilms grown overnight with Esp452-AF647. Bacteria were stained with Syto-13 (green) and proteins were labeled with AF647 (red). The first image was taken at the interface of the biofilm with the slide and each stack is 1.2 μm higher, progressing up to the top of the biofilm at the biofilm-media interface.
(PDF)

## Acknowledgments

We thank the late C. Spiegelman for his help with statistical analyses, N. Devaraj for the use of his lab's Tecan plate reader, and P. Kolesiński for his assistance analyzing crystallographic data. ESI mass spectrometry was supported by NIH S10RR25636. The Protein-Glycan Interaction Resource of the CFG and the National Center for Functional Glycomics (NCFG) at Beth Israel Deaconess Medical Center, Harvard Medical School was supported by NIH grants

P41GM103694 and R24GM137763. Beamline 12.3.1 and 5.0.2 of the Advanced Light Source, a U.S. DOE Office of Science User Facility under Contract No. DE-AC02-05CH11231, is supported in part by the ALS-ENABLE program funded by the NIH grant P30 GM124169-01. This research used resources of the Advanced Photon Source, a U.S. Department of Energy (DOE) Office of Science user facility operated for the DOE Office of Science by Argonne National Laboratory under Contract No. DE-AC02-06CH11357.

## Author Contributions

**Conceptualization:** Lindsey Spiegelman, Partho Ghosh.

**Formal analysis:** Lindsey Spiegelman, Greg L. Hura, Susan E. Tsutakawa, Partho Ghosh.

**Funding acquisition:** Partho Ghosh.

**Investigation:** Lindsey Spiegelman, Adrian Bahn-Suh, Elizabeth T. Montaño, Ling Zhang, Greg L. Hura, Kathryn A. Patras, Amit Kumar.

**Methodology:** Lindsey Spiegelman.

**Project administration:** Partho Ghosh.

**Supervision:** F. Akif Tezcan, Victor Nizet, Susan E. Tsutakawa, Partho Ghosh.

**Visualization:** Lindsey Spiegelman, Greg L. Hura, Kathryn A. Patras, Susan E. Tsutakawa.

**Writing – original draft:** Lindsey Spiegelman, Elizabeth T. Montaño, Greg L. Hura, Kathryn A. Patras, Susan E. Tsutakawa, Partho Ghosh.

**Writing – review & editing:** Lindsey Spiegelman, Adrian Bahn-Suh, Elizabeth T. Montaño, Greg L. Hura, Kathryn A. Patras, F. Akif Tezcan, Victor Nizet, Susan E. Tsutakawa, Partho Ghosh.

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
