## [Decision Letter · Decision Letter 0]

15 Jul 2022

Dear Dr. Ghosh,

Thank you very much for submitting your manuscript "Strengthening of enterococcal biofilms by Esp" for consideration at PLOS Pathogens. As with all papers reviewed by the journal, your manuscript was reviewed by members of the editorial board and by several independent reviewers. The reviewers appreciated the attention to an important topic. Based on the reviews, we are likely to accept this manuscript for publication, providing that you modify the manuscript according to the review recommendations. Reviewer 1 in particular presents a number of ideas that would strengthen the manuscript. 

Sincerely,

Theresa M. Koehler

Associate Editor

PLOS Pathogens

Guy TRAN VAN NHIEU

Section Editor

PLOS Pathogens

Kasturi Haldar

Editor-in-Chief

PLOS Pathogens

orcid.org/0000-0001-5065-158X

Michael Malim

Editor-in-Chief

PLOS Pathogens

orcid.org/0000-0002-7699-2064

Reviewer Comments (if any, and for reference):

Reviewer's Responses to Questions

**Part I - Summary**

Reviewer #1: In the manuscript, “Strengthening of enterococcal biofilms by Esp,” the authors solve the structure of the N-terminal region of Esp and elucidate the mechanism to which it contributes to biofilm formation as well as the environmental conditions required – low pH. First, the authors solve the structure of the N-terminal non-repeat region of Esp (Esp452) and observed two globular domains with resemblance to MSCRAMMS, a family of proteins involved in bacterial adhesion. When the protein is added to developing biofilms, it does not help adhesion, but does help strengthen the developing biofilm such that it is more wash resistant. In contrast, the other half of the N-terminus (Esp452-743) does not help, and inclusion of this half (Esp743) also does not have a strengthening effect at the pH used, 4.5. However, the authors discover they if they lower the pH by buffering, or by including more glucose for further acidification by fermentation, Esp743 can strengthen biofilms. The authors also show that Esp452 binds Thioflavin T, indicative of amyloid formation, and co-localizes with the biofilm. Finally, the authors show that an even shorter version EspDDDK (50-256) has the same properties as Esp452. The DDDK sequence represents a protease cleavage site, and the authors speculate host proteases may cleave Esp to cause a biofilm strengthening effect.

Overall, the study is well executed, and the conclusions well supported. The manuscript is a significant and valuable contribution to the field of bacterial biofilm formation in that it elucidates the amyloid-forming mechanism by which Esp strengthens biofilms. By discovering that this process requires low pH, the manuscript very much helps explain the contradictory data on Esp in previous publications – some observed Esp to positively effect biofilm formation and some did not. While I anticipate this work being well-received by the Enterococcus community, I do have the following suggestions to further improve and clarify the manuscript.

Reviewer #2: This manuscript by Spiegelman et al. is an interesting study on the role of enterococcal surface protein (Esp) in biofilms produced by E. faecalis. This is an excellent, thorough study that combines structural biology, biochemistry, and genetics. The authors characterized the N-terminal region of Esp, showing that this protein belonged to the MSCRAMM family of surface proteins. They tested a number of ligands for Esp binding (including fibronectin and glycans) and also demonstrated that Esp is likely not important in the urinary tract using an animal model. A majority of the paper focused on the biofilm-strengthening role of the Esp452 variant and the requirement of reduced pH for this strengthening to occur. Addition of Esp452 to E. faecalis cultures made the biofilms more resistant to washing and perturbation by DNase. Importantly, exposure to low pH resulted in aggregation of Esp452, and the authors show a mechanism for generation of a similar fragment through host protease activity. This work also provides important insight into the role of glucose in E. faecalis biofilm formation (given the differing reports on biofilm formation and genes required for biofilm formation in various culture conditions in the literature) as well as interpretation regarding expression levels of esp from a plasmid in previous work.

Together, I think the experiments support the conclusions regarding the role of Esp in biofilm strengthening at low pH in E. faecalis biofilms. The manuscript is well-cited, and the authors do an excellent job of taking outstanding questions in the E. faecalis biofilm literature and answering them using their biochemical approaches. Additionally, the authors provide clear rationale for their experimental choices, such as evaluating biofilms by ATP content instead of crystal violet staining.

**Part II – Major Issues: Key Experiments Required for Acceptance**

Reviewer #1: (No Response)

Reviewer #2: No key experiments required for acceptance.

**Part III – Minor Issues: Editorial and Data Presentation Modifications**

Reviewer #1: 1) Figure 1A: It is not clear what the difference is between Esp452 and EspDDDK until the very end of the Results section though it is mentioned earlier in the manuscript including in the Introduction. It would improve the clarity to mention what EspDDK is earlier and include it in it in Figure 1A.

2) Figure 3B: The legend picture in Figure 3B is not right – a gray bar is depicted where there should be white bar.

3) Figure 4C: The biofilm co-localization by Esp452 and Esp743 in Figure 4C shows just one image field is shown that could have been cherry-picked. A Mander’s overlap coefficient is mentioned, but there is no description. Can a more thorough description of what this is and how the data was analyzed be included in the results? Were multiple independent images analyzed? Can a Mander’s overlap coefficient be calculated for multiple fields and then averages calculated to assess how consistent the strength/weakness of the overlap is?

4) Comment/question: Overexpression of the protein can cause the strengthening effect on the biofilms to occur at higher pHs. Do you think it is possible some strains of Enterococcus express more Esp than others contributing to natural strain variation and niche adaptation?

5) Too much data is in the supplement, much of which is important to the story. Not one main figure has more than 4 panels and more of the relevant data could be added to the main figures to make the paper easier to read. I suggest moving S5 (the actual visualization of the differences in the biofilms strengthens the story as the ATP measurements are not standard to the field) S12, S15, S17 and S18.

Reviewer #2: 1) Could the authors specify which Corning 96-well plates were used (3595, 3596, etc) in the Methods section? The plate material and any surface treatments result in changes in hydrophobicity, which can impact biofilm formation.

2) Figure 2A shows a low percentage of signal coming from planktonic vs biofilm cells. This is slightly surprising given the high number of cells in the planktonic fraction (typically around 10^9 CFU/mL). Does this mean that the CFU of biofilm cells > planktonic cells? Or perhaps that biofilm cells are more amenable to lysis with the CellTiter-Glo reagent?

3) Line 437 – “Esp452 bound ThT maximally at pH 4.5 (Fig. 3D)” – the average values for Esp452 at pH 4.2 and 4.3 appear to be greater than the value at 4.5 (although not by much)

PLOS authors have the option to publish the peer review history of their article (what does this mean?). If published, this will include your full peer review and any attached files.

Reviewer #1: No

Reviewer #2: No

Figure Files:

Data Requirements:

Reproducibility:

References:

---

## [Editor Report · Decision Letter 1]

22 Aug 2022

Dear Dr. Ghosh,

We are pleased to inform you that your manuscript 'Strengthening of enterococcal biofilms by Esp' has been provisionally accepted for publication in PLOS Pathogens.

Best regards,

Theresa M. Koehler

Associate Editor

PLOS Pathogens

Guy TRAN VAN NHIEU

Section Editor

PLOS Pathogens

Kasturi Haldar

Editor-in-Chief

PLOS Pathogens

orcid.org/0000-0001-5065-158X

Michael Malim

Editor-in-Chief

PLOS Pathogens

orcid.org/0000-0002-7699-2064
---

## [Editor Report · Acceptance letter]

8 Sep 2022

Dear Dr. Ghosh,

We are delighted to inform you that your manuscript, "Strengthening of enterococcal biofilms by Esp," has been formally accepted for publication in PLOS Pathogens.

Best regards,

Kasturi Haldar

Editor-in-Chief

PLOS Pathogens

orcid.org/0000-0001-5065-158X

Michael Malim

Editor-in-Chief

PLOS Pathogens

orcid.org/0000-0002-7699-2064